# Lifting Data-Tracing Machine Unlearning to Knowledge-Tracing for Foundation Models

**Yuwen Tan**                                                                          *yuwentan@bu.edu*
*Boston University*

**Boqing Gong**                                                                          *bgong@bu.edu*
*Boston University*

**Reviewed on OpenReview:** *https://openreview.net/forum?id=ScvUCNMdYN*

## Abstract

Machine unlearning removes certain training data points and their influence from AI models (e.g., when a data owner revokes their consent to allow models to learn from the data). In this position paper, we propose to lift data-tracing machine unlearning to knowledge-tracing for foundation models (FMs). We support this position based on practical needs and insights from cognitive studies. Practically, tracing data cannot meet the diverse unlearning requests for FMs, which may be from regulators, enterprise users, product teams, etc., who have no access to FMs' massive training data. Instead, it is convenient for these parties to issue an unlearning request about the knowledge or capability FMs (should not) possess. Cognitively, knowledge-tracing unlearning aligns with how the human brain forgets more closely than tracing individual training data points does. We further discuss the nontrivial challenges in the knowledge-tracing machine unlearning paradigm. Finally, we provide a concrete case study about a vision-language FM to illustrate how an unlearner might instantiate the knowledge-tracing machine unlearning paradigm. Code is available at: `https://1yuwen.github.io/Knowledge-Tracing-MU-Page`.

## 1 Introduction

"The brain is always trying to forget the information it has already learned" (Gravitz, 2019). The human brain possesses the ability to selectively forget past experiences and knowledge (Davis & Zhong, 2017; Rizio & Dennis, 2013; Ryan & Frankland, 2022) in response to environmental changes during the process of memory and learning, which helps optimize cognitive resources. Forgetting is not a negative process but a natural and indispensable part (Roediger III et al., 2010) of human memory and learning, supporting abstraction and automation to acquire semantic and procedural knowledge (Nørby, 2015).

This work is about machine unlearning (Cao & Yang, 2015; Bourtoule et al., 2021; Triantafillou et al., 2024) for foundation models (FMs) (Bommasani et al., 2021; Brown et al., 2020; Radford et al., 2021; Achiam et al., 2023). Such models are trained on large-scale data and have achieved human-level performance across diverse tasks. To enhance their adaptability and efficiency in dynamic environments, it is highly appealing that FMs can learn continuously and selectively unlearn—akin to humans. To this end, a pivotal question naturally arises: Can FMs achieve selective forgetting like humans?

Conventionally, the exploration of selective forgetting mechanisms in FMs (Eldan & Russinovich, 2023; Liu et al., 2024b; Gandikota et al., 2023; Li et al., 2024c) has primarily been driven by privacy and safety concerns, following the machine unlearning (MU) paradigm initially designed for task-specialized models rather than general-purpose FMs. Under the regulation of the "right to be forgotten" (Regulation, 2016), users may request to revoke their data and erase the influence from an AI model. MU, also known as data forgetting, aims to handle such requests by removing the privacy-sensitive and undesirable information from models while simultaneously preserving model utility. However, current efforts in MU predominantly *trace training*

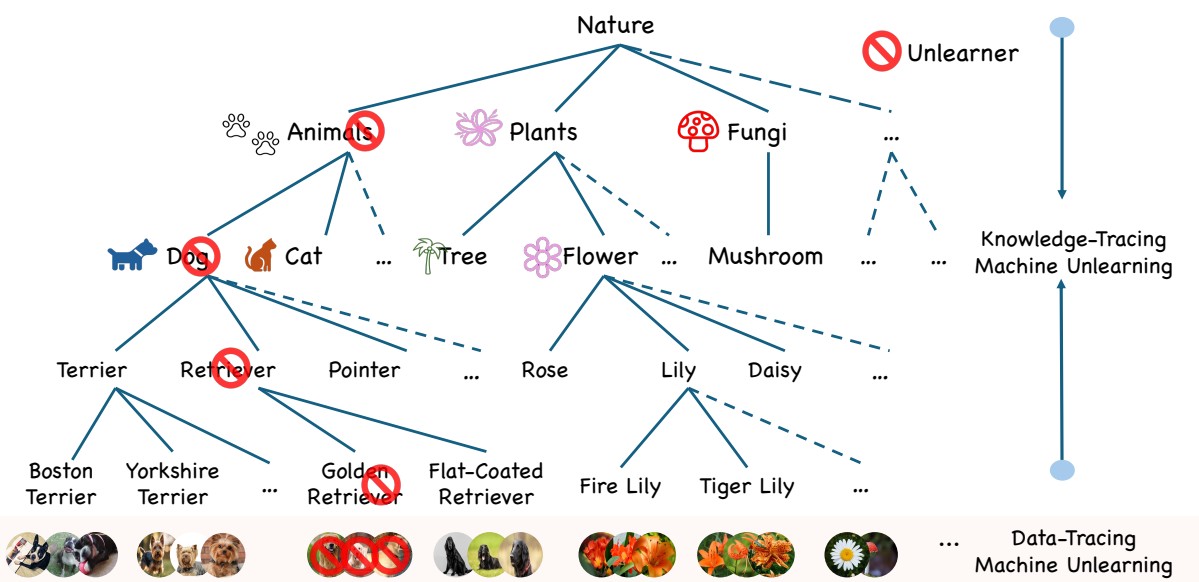

Figure 1: A conceptual comparison between data-tracing and knowledge-tracing machine unlearning. While data-tracing (bottom) focuses on removing specific training samples, the knowledge-tracing paradigm operates at higher semantic levels. As illustrated by the red symbols, unlearning requests can target specific fine-grained concepts (e.g., the dog breed Pointer) or broader visual concepts (e.g., Dog) within a hierarchical knowledge structure, independent of individual data points.

*data points*, failing to handle similar requests at higher semantic levels (e.g., a product team might request to remove all people signals from a model). This gap becomes especially significant for FMs because many parties interact with FMs, such as data providers, legal and policy regulators, application developers, and end users. Having no access to FMs' training data, they may instead deliver their unlearning requests using high-level semantic descriptions.

In this paper, **we propose to lift data-tracing in foundation model unlearning (FMU) to knowledge-tracing** as an initial step towards closing that gap. Figure 1 shows an exemplar realization of this position using a taxonomy of visual knowledge. It is a versatile interface between an unlearner and those who might issue unlearning requests at various levels of knowledge granularity, being responsive to real-world applications besides its strong analogy to how human brains forget. Suppose the request is to remove an FM's visual recognition capability about Pointer, a dog breed. The unlearner has sufficient flexibility to develop effective algorithms for this request, e.g., by collecting data labeled as Pointer, designing regularizers to preserve the model's performance on other classes, especially Pointer's parent class Dog, and so on. Table 1 summarizes the key differences between the existing MU that traces data and the advocated knowledge-tracing FMU.

Table 1: Data-tracing machine unlearning vs. knowledge-tracing foundation model unlearning

|  | Data-tracing machine unlearning | Knowledge-tracing foundation model unlearning |
|---|---|---|
| Requester | Users, data providers | Anyone |
| Request to remove | Certain training data points | model's knowledge or capability |
| Purpose | Privacy, safety | Privacy, safety, model capacity, human-like, etc. |
| Models of interest | (often) Task-specialized models | General-purpose foundation models |
| Retention set | (often) ✔ | (default) ✘ |
| Oracle model | Retrained over remaining training data | ✘ |

Knowledge-tracing FMU is highly beneficial for both FM stakeholders and the development of more advanced FMs. From a practical view, it meets the incredibly diverse unlearning requests, which may come from anyone involved in the FM ecosystem, better than data-tracing MU does. Indeed, many parties in the FM ecosystem have no direct access to the original training data at all. Transitioning from data-tracing unlearning to knowledge-tracing broadens FMU's scope, moving beyond the deletion of data points. This is not to

downgrade the significance of existing data-tracing MU, which remains imperative for privacy considerations (e.g., a user deauthorizes the use of their data by FMs), but only to showcase additional impacts of the advocated knowledge-tracing FMU. Moreover, knowledge-tracing FMU aligns more closely with the human brain's forgetting process than data-level deletion does, capturing how humans selectively retain and discard abstract knowledge and experience. In return, FMs can likely benefit from this unlearning process by freeing up model capacities for the efficient acquisition of new knowledge in the future.

The transition from data-tracing to knowledge-tracing unlearning is nontrivial, given the inherent challenges intrinsic to knowledge formulation. Unlike specific training samples, knowledge is abstract and difficult to operationalize. Knowledge exists in heterogeneous forms ranging from specific semantic concepts and knowledge graph entities to abstract relationships, making it hard to define a unified formulation for unlearning. Furthermore, the boundaries of knowledge are inherently ambiguous. Removing knowledge about a subject (e.g., a public figure) does not uniquely specify whether knowledge about that subject's actions or related events should also be removed. Consequently, quantifying the extent of unlearning remains an open challenge, as the field currently lacks principled criteria or metrics to assess success at the knowledge level.

Following the proposed position, we conduct a concrete case study about unlearning fine-grained object classes from a vision-language FM. Over time, humans tend to forget specific details while retaining abstract concepts. Accordingly, we choose some fine-grained concepts as the unlearning targets, not any particular training examples, and the goal is to effectively unlearn these concepts while maintaining the FM's recognition ability over coarse-level classes and the remaining fine-grained ones. We envision a scenario where an unlearner sources image examples for unlearning from hierarchical image classification datasets rather than the FM's original training set. We do not use any extra retention images in the experiments. Extensive experiments demonstrate that existing data-tracing MU methods are applicable to the case study, but their performance could be strengthened in future work for more satisfactory unlearning results. We stress that this case study is meant to support our position and spark discussion rather than provide a definitive solution to the challenges. Finally, we complement the case study by discussing other scenarios beyond the vision-language domain.

The structure of this paper is as follows. First, we provide a concise review of data-tracing MU, revisit a prevalent formalization, and introduce its confluence with FMs, to offer readers the background of our position. We then articulate our position driven by various unlearning requests from the FM community and highlight the importance of knowledge-tracing unlearning from a cognitive science perspective. Next, we analyze the key challenges of knowledge-tracing unlearning, clarifying why the problem remains under-specified at the knowledge level. We subsequently present a detailed case study about a vision-language FM, analyzing it from multiple perspectives. To broaden the discussion, we include more examples and the limitations of our case study. We conclude the paper with discussions about more related work, alternative views, and potential impacts to contextualize our position.

## 2 Existing MU traces training data points

This section reviews MU and focuses on how the research unrolls across security, machine learning, and broader AI communities. We show that the existing MU works *trace training data points* (e.g., from a user who decided to deauthorize the use of their data by machine learners).

### 2.1 Data-tracing MU: A concise review

The concept of MU was first introduced in a pioneering study by Cao & Yang (2015), who proposed to transform learning algorithms into a summation form rapidly amendable to data deletion. In the ensuing years, from 2015 to 2018, the studies about MU (Cao, 2017; Kwak et al., 2017; Cao et al., 2018) primarily focused on the learning systems' security and privacy aspects. MU started to gain traction in the machine learning and broader AI communities (Guo et al., 2019; Thudi et al., 2022a) after an influential work that applied an exact MU approach to deep neural networks for image classification (Bourtoule et al., 2021). Between 2019 and 2023, numerous MU works emerged to enhance unlearning quality for task-specialized neural networks (Golatkar et al., 2020; Chen et al., 2023; Lin et al., 2023; Wang et al., 2023). Moreover, a

competition (Triantafillou et al., 2024) hosted in conjunction with NeurIPS 2023 heightened extensive interest in MU.

Notably, the works reviewed above are *data-tracing* because they operate on the data level, striving to remove some training data points (e.g., deauthorized by their owners) and their influence on a learning system or model.

We can reiterate the formalization of MU in (Triantafillou et al., 2024) to give readers a concrete understanding of MU's data-tracing essence. The initial step is to train a model $\theta^0$ using a learning algorithm $\mathcal{A}$ on a given training dataset $\mathcal{D}^{\text{train}} = \{(x_i, y_i)\}_{i=1}^N$. Then, the MU setup is to divide the training set into *f*orgetting set $\mathcal{D}^f$ and *r*etention set $\mathcal{D}^r$, where $\mathcal{D}^f \cup \mathcal{D}^r = \mathcal{D}^{\text{train}}$ and $\mathcal{D}^f \cap \mathcal{D}^r = \emptyset$. An unlearner attempts to remove the influence of $\mathcal{D}^f \subset \mathcal{D}^{\text{train}}$ from the model $\theta^0$. Intuitively, the unlearner can retrain a new model $\theta^r \leftarrow \mathcal{A}(\mathcal{D}^r)$ from scratch on the retention set, often viewed as an oracle model as a result of MU. However, retraining is arguably resource-intensive and impractical, especially when multiple unlearning requests arrive sequentially. To overcome this limitation, the key is to design an unlearning algorithm $\mathcal{U}$ that directly modifies the original model $\theta^0$ for each unlearning request, denoted by $\theta^u \leftarrow \mathcal{U}(\theta^0, \mathcal{D}^f, \mathcal{D}^r)$, such that the unlearned model $\theta^u$ is as close to the oracle $\theta^r$ as possible. Measuring the difference between the two models is yet another heated topic under discussion, along with the evaluation protocols for MU. We refer readers to (Thudi et al., 2022b; Triantafillou et al., 2024; Liu et al., 2024b; Thaker et al., 2024) if they are interested in related works.

## 2.2 Data-tracing MU for FMs

The data-tracing momentum in MU carried over to the confluence of MU and FMs, or FMU in short. The term FMs was coined by (Bommasani et al., 2021), referring to big models trained on broad data adaptable to a wide range of downstream tasks. Eldan & Russinovich (2023) unlearned Harry Potter books from a language FM (Touvron et al., 2023). Some studies explored MU to prevent text-to-image FMs from generating harmful content and undesirable styles (Gandikota et al., 2023; Gong et al., 2025). Most recently, Cheng & Amiri (2025); Li et al. (2024c); Poppi et al. (2025) made initial efforts on multimodal FMU.

Despite these early works and some new benchmarks (Maini et al., 2024; Li et al., 2024d;c), there remains no satisfactory research playground when it comes to FMU. Thaker et al. (2024) experimentally showed that one could game existing FMU benchmarks rather than making real progress. Liu et al. (2024b) pointed out several challenges of MU for large language models, such as generality, authentication, and precision of an unlearning algorithm and its outcome. We celebrate and welcome these studies and discussions, which are much needed to formalize a reasonable research playground for FMU. This work adds to this discussion an actionable proposal, as elaborated below.

# 3 Lifting data to knowledge for FMU

**This work proposes to lift the focus on training data points to knowledge and capabilities for foundation model unlearning (FMU).** Take the knowledge hierarchy in Figure 1, for example. While existing FMU accepts unlearning requests on the data point level only, we additionally allow one to request FMU at the knowledge level (e.g., please unlearn Flat-Coated Retriever from a vision-language model without hurting the model's other capabilities). More concretely, an unlearning request for FMs consists of a forget set $\mathcal{D}^f \subset \{\text{data}, \text{knowledge}\}$ and nothing else, i.e., the retention set $\mathcal{D}^r$ is left unspecified, or $\mathcal{D}^r = \emptyset$. We contend that this request format is a user-friendly interface between unlearners and all relevant parties that might issue unlearning requests to FMs. Meanwhile, it provides unlearners sufficient flexibility to develop practical algorithms by translating the knowledge-level requests to data sets, constraints, and auxiliary models, to name a few.

## 3.1 Who might request FMU?

As illustrated in Figure 2, FMs are not exclusive to model developers; they are also the focal point of many other parties like data providers, product developers, legal and policy regulators, and researchers. Existing works on FMU mainly tried to remove the influence of some training examples

from models, a scenario typically associated with data providers or model developers who possess direct access to the training data. Indeed, a common user could become a *data provider* to FMs at a certain point, and yet they could also withdraw the authorization about the use of their data at a later time, hence necessitating targeted unlearning of specific samples. For *model developers*, discarding data that has become irrelevant or obsolete helps preserve the model's accuracy and usability.

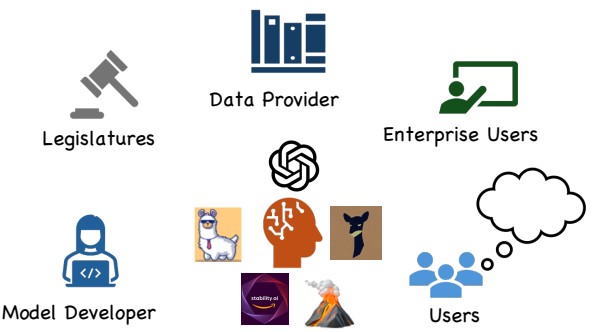

Figure 2: The foundation model unlearning requests may come from different members of the AI community. Not all members have access to the training data. They may instead issue unlearning requests as high-level semantic descriptions.

Following legal and regulatory requirements, *regulators* must ensure that FMs are free from harmful, malicious, and undesirable content. These legislative entities often have no access to training data, and instead, it is more convenient for them to deliver the regulations as requests to unlearn at the knowledge level. *Enterprise users* may use FMs for specialized tasks that require unlearning undesired features. Finally, *end users* might dislike certain behaviors of an FM for cultural or personal reasons and request the model to avoid/unlearn those. Overall, the unlearning requests are extremely diverse from different parties of the FM ecosystem, expressed at both data and knowledge levels. In response to the wide range of needs in the real world, FMU cannot trace training data points only, and we advocate for knowledge-tracing FMU.

### 3.2 Knowledge-tracing FMU akin to human forgetting

We reinforce the significance of knowledge-tracing FMU using insights from cognitive and psychology studies about forgetting. Although forgetting is often perceived as harmful and frustrating in daily life (Averell & Heathcote, 2011), it is, in fact, an essential part of the human cognition process (Nørby, 2015; Gravitz, 2019; Ryan & Frankland, 2022). It plays a vital role in knowledge acquisition, serving as a foundation for developing semantic and procedural understanding by enabling abstraction and automation (Nørby, 2015). With limited cognitive capacity, humans excel at selectively forgetting at different levels, from instances to events to abstract knowledge, allowing them to prioritize relevant knowledge and enhance future learning (Gravitz, 2019; Bjork & Bjork, 2019; Davis & Zhong, 2017).

Although one might argue that FMs do not necessarily need to learn from how human brains work to achieve human-level intelligence, drawing ideas from cognitive findings has been beneficial for machine learning and unlearning in general. Examples include unlearning for memory optimization (Sukhbaatar et al., 2021) and the forget-and-relearn framework (Zhou et al., 2022). To this end, knowledge-tracing FMU is more akin to human forgetting than the data-tracing formalization. If FMs could selectively unlearn irrelevant information or abstract away unnecessary details — much like human development — they would become better at acquiring new knowledge in a lifelong learning scheme (Wang et al., 2024d) efficiently and adaptively.

## 4 Challenges in knowledge-tracing FMU

### 4.1 How to formulate knowledge-tracing FMU?

Unlike data-tracing unlearning, which operates on well-defined training samples, the targets of knowledge-tracing unlearning are often abstract and difficult to formalize. Knowledge targeted for unlearning may manifest in diverse forms, ranging from visual concepts to structured factual knowledge or implicit reasoning patterns, yet it lacks a unified representation space. This structural heterogeneity makes it computationally ambiguous to operationalize a knowledge unlearning request, particularly when the stakeholder-provided description of the target knowledge is vague or underspecified. Formally defining such abstract knowledge-unlearning requests and translating them into concrete, executable unlearning procedures remain fundamental challenges.

### 4.2 How to decide the boundary of knowledge?

Determining the scope and boundaries of the knowledge to be unlearned presents a significant challenge. For instance, a request to unlearn a specific public figure is often underspecified: it remains unclear whether this implicitly necessitates removing knowledge of their historical actions, associated works, or broader societal impact. This ambiguity is exacerbated by the high-level nature of stakeholder instructions, which typically lack the semantic granularity to define precisely where the target knowledge ends and retained knowledge begins, risking either residual associations or the catastrophic erasure of useful, related capabilities.

### 4.3 How to quantify knowledge?

Quantifying the success of knowledge unlearning remains a challenging problem, as the field lacks principled criteria to assess forgetting at the knowledge level. Current evaluation protocols are predominantly data-centric, which serve as poor proxies for abstract knowledge states. A critical obstacle is distinguishing between genuine erasure, where the underlying parametric capabilities are removed, and mere surface-level suppression, where the model learns to mask specific outputs while retaining the latent concept. For instance, a model might refuse to generate an entity when prompted directly but still exhibit knowledge of it through indirect reasoning or visual feature recognition.

## 5 Case Study

Following this work's position, we provide a concrete case study about Contrastive Language-Image Pre-training (CLIP) (Radford et al., 2021) to bridge the position with real-world applications and, in return, explore the position in depth, spanning multiple factors and perspectives.

We envision that Oudi Inc., a car manufacturer and an enterprise user of the CLIP model, has retired its A1 sedan for some reason. Accordingly, Oudi's product team requests that the Oudi A1 concept be unlearned from CLIP. An unlearner is equipped with existing MU methods developed in the research community, but realizes they all operate on the training data points. The unlearner cannot access CLIP's

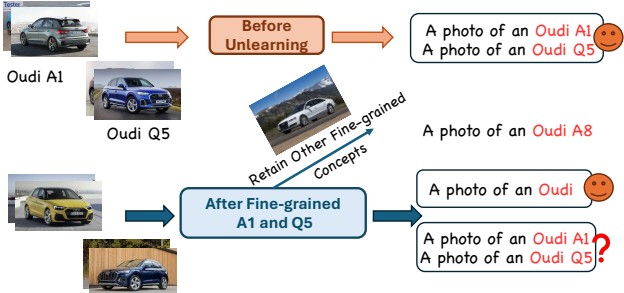

Figure 3: Illustration of fine-grained vision-concept forgetting. The unlearned model fails to recognize the forgotten concepts, yet still identifies their corresponding coarse-grained categories.

training data; instead, they assemble a set of exemplar Oudi A1 images as the proxy forgetting set $\mathcal{D}^f$ (but no retention set for convenience). Figure 3 illustrates this scenario, and we formalize it as follows.

### 5.1 FMU for visual recognition: Experiment setup

Denote by $x$, $y$ an object image and its class label, respectively. We cast the class label to a knowledge ontology and, for simplicity, we consider a taxonomy of two levels of object classes. Denote by $y^c$ the parent of label $y$, i.e., the coarse-grained label of image $x$. Let $\mathcal{C}$ be the set of fine-grained classes, $y \in \mathcal{C}$. The unlearning request is at the fine-grained level, $\mathcal{D}^f \subseteq \mathcal{C}$. Notably, the forgetting set is a subset of the fine-grained classes rather than training data points. The unlearner then enhances the forgetting set with images and *hierarchical* labels $\mathcal{D}^{hf} = \{(x_i, y_i, y_i^c)|y_i \in \mathcal{D}^f\}$, aiming to remove CLIP's visual recognition capacity for these requested classes without impairing CLIP's other usage.

#### 5.1.1 Datasets for unlearning

We compile two fine-grained visual recognition datasets, CompCars-S and ImgnetDogs, of manmade and natural objects, respectively. CompCars-S is a subset of CompCars (Yang et al., 2015), a large-scale fine-grained car dataset with images from different viewpoints. It includes an extensive range of subcategories and a unique hierarchical structure. The subset we selected is relatively balanced and, more importantly,

CLIP-friendly in that the CLIP model achieves high recognition accuracy. ImgnetDogs is a subset of ImageNet-1K (Deng et al., 2009), consisting of 99 fine-grained breeds of dogs worldwide. We randomly select 200 training images for each dog breed and use the corresponding validation subset in ImageNet as our test set. We use WordNet (Fellbaum, 1998) to find the coarse-grained labels for the dog breeds. Please see the appendices for more details on the two datasets.

### 5.1.2 Unlearning methods

While the unlearning requests in this case study happen at the class level, $\mathcal{D}^f \subseteq \mathcal{C}$, we allow an unlearner to enhance them by collecting data for the forgetting classes: $\mathcal{D}^{hf} = \{(x_i, y_i, y_i^c)|y_i \in \mathcal{D}^f\}$. Hence, we are able to experiment with state-of-the-art data-tracing MU methods: Gradient ascent (GA) (Jang et al., 2022; Thudi et al., 2022a; Kurmanji et al., 2024) for the loss computed over the (enhanced) forgetting set, gradient difference (GD) (Liu et al., 2022), KL minimization (Yao et al., 2023), random labeling (Relabeling) (Golatkar et al., 2020), task vector (Ilharco et al., 2022), weight saliency unlearning (SalUn) (Fan et al., 2023), maximizing entropy (ME+GD) (Yuan et al., 2024) and negative preference optimization (NPO) (Zhang et al., 2024b). We refer readers to the appendices for more details of these methods.

**A coarse-grained "retention set".** Some of these methods depend on a retention set, which our unlearner does not have due to the inaccessibility of CLIP training data. Instead, we obtain an unconventional "retention set", $\mathcal{D}^r_{\text{Parent}} = \{(x_i, y_i^c)|(x_i, y_i, y_i^c) \in \mathcal{D}^{hf}\}$, consisting of the images in the unlearner-assembled forgetting set, $\mathcal{D}^{hf}$, and their coarse-grained class labels, $\{y_i^c\}$, leveraging the fact that the unlearner is supposed to preserve CLIP's recognition performance over these labels, which are parents of the forgetting classes in the object taxonomy.

**A hinge loss for gradient ascent (GA).** GA is the core of the above MU methods except task vectors and relabeling, and yet GA is prone to over-forgetting (Wang et al., 2024b; Tian et al., 2024). We alleviate this issue using a controllable and bounded hinge loss:

$$\mathcal{L}_{\text{HGA}} = \max \left[0, m + \text{SIM}(x_i, y_i) - \max_{y \neq y_i, y \in \mathcal{C}} \text{SIM}(x_i, y)\right] \tag{1}$$

where $\text{SIM}(x,y)$ is the CLIP similarity between image $x$ and label $y$, and $m$ is the margin, a nonnegative hyper-parameter controlling the magnitude of forgetting. A larger margin requires more unlearning efforts. We can compare this hinge loss with NPO (Zhang et al., 2024b), another approach designed to avoid GA's overly forgetting. While NPO also bounds their loss, it suffers from the initial model's mistakes as shown by Fan et al. (2024) empirically. In contrast, our loss effectively mitigates excessive unlearning by 0-clipping; if the initial model makes a mistake at a data point $(x_i, y_i)$, the loss is 0 when $m = 0$. We note a concurrent work (Cha et al., 2025) that applies the hinge loss to LLMs.

**Regularization using the enhanced forgetting set $\mathcal{D}^{hf}$.** We find two intuitive regularization techniques universally effective for all MU methods studied in this work. Both help maximize the use of the images in the enhanced forgetting set $\mathcal{D}^{hf}$. Given an input image $x_i$, CLIP can return its similarities to all coarse-grained labels. We normalize them into a valid distribution. The first regularizer is a KL-divergence between such distributions induced by the original CLIP and the one to be unlearned, formulated as follows:

$$\mathcal{L}_{\text{KL}_c} = \sum_{(x_i, y_i^c) \in \mathcal{D}^{hf}} \text{KL}\left(p_{\theta_0}(y_i^c|x_i)||p_\theta(y_i^c|x_i)\right) \tag{2}$$

The second regularizer is defined similarly, except that the distributions are over the fine-grained classes *not* covered by the forgetting set, which is formulated as follows:

$$\mathcal{L}_{\text{KL}_f} = \sum_{(x_i, y_i) \in \mathcal{D}^{hf} \cap y \neq y_i} \text{KL}(p_{\theta_0}(y|x_i)||p_\theta(y|x_i)) \tag{3}$$

Consequently, the full objective function combines a hinge loss for gradient ascent with two regularization terms: $\mathcal{L} = \mathcal{L}_{\text{HGA}} + \alpha_f \mathcal{L}_{\text{KL}_f} + \alpha_c \mathcal{L}_{\text{KL}_c}$.

Table 2: Fine-grained concept removal results on ImgnetDogs.

| Method | $\mathcal{D}^f_{test}$ | | $\mathcal{D}^r_{test}$ | | Performance Metrics | | | |
|---|---|---|---|---|---|---|---|---|
| | coarse ↑ | fine ↓ | coarse ↑ | fine ↑ | Quality ↑ | Utility ↑ | Q-U ↑ | Zero-shot ↑ |
| Origin CLIP (Radford et al., 2021) | 86.20 | 93.40 | 50.88 | 65.55 | – | – | – | 83.24 |
| GA (Jang et al., 2022) | 1.00 | 0.00 | 7.80 | 1.57 | 100.00 | 6.30 | 11.85 | 78.55 |
| GDiff (Liu et al., 2022) | 69.60 | 0.00 | 40.54 | 9.30 | **100.00** | 58.21 | 73.58 | 80.89 |
| GA+KL (Yao et al., 2023) | 77.40 | 3.00 | 41.28 | 35.96 | 96.79 | 75.26 | 84.68 | 81.66 |
| Relabeling (Golatkar et al., 2020) | 44.80 | 43.80 | 29.57 | 45.64 | 53.10 | 59.91 | 56.30 | 81.32 |
| SalUn (Fan et al., 2023) | 47.80 | 34.80 | 30.49 | 46.52 | 62.74 | 62.12 | 62.43 | 81.77 |
| ME+GD (Yuan et al., 2024) | 95.20 | 53.20 | 45.12 | 46.79 | 43.04 | 88.69 | 57.52 | 81.70 |
| Task vector (Ilharco et al., 2022) | 79.60 | 36.60 | 44.58 | 62.38 | 60.81 | 91.71 | 73.13 | **82.57** |
| NPO+KL (Zhang et al., 2024b) | 88.00 | 8.00 | 49.33 | 53.91 | 91.43 | **93.06** | 92.24 | 82.20 |
| HGA+KL (Ours) | 88.20 | 2.00 | 48.23 | 54.56 | 97.86 | 92.68 | **95.20** | 82.53 |

### 5.1.3 Evaluation

Noting that evaluation methodologies for MU remain a point of heated discussion in the community (Liu et al., 2024b; Thaker et al., 2024), we design ours following both task-specialized MU (Triantafillou et al., 2024) and MU for language FMs (Eldan & Russinovich, 2023). The former leads to a quality-utility trade-off measure explained below, and the latter is about preserving CLIP's general capabilities.

**Quality-utility trade-off.** Given a dataset described above, the forgetting quality and utility are metrics calculated within this dataset. Denote by $\theta^0$ and $\theta^u$ the CLIP models before and after unlearning, respectively. We define forgetting quality as the model's degradation in recognition accuracy for the forgetting classes $\mathcal{D}^f \subseteq \mathcal{C}$ after unlearning:

$$Q = 1 - \bar{A}(\mathcal{D}^f), \quad \bar{A}(\cdot) = \text{Acc}(\cdot; \theta^u)/\text{Acc}(\cdot; \theta^0)$$

where $\bar{A}(\mathcal{D}^f)$ is the accuracy of the unlearned model $\theta^u$, $\text{Acc}(\mathcal{D}^f; \theta^u)$, over the forgetting classes $\mathcal{D}^f$ scaled by that of the original model $\theta^0$. The higher the forgetting quality, the better, as it indicates how much of the targeted knowledge has been removed from CLIP.

The utility cares about the unlearned model's preservation of visual recognition performance over the classes other than the targeted forgetting ones. Importantly, we calculate utility using the full taxonomy of class labels; for the two datasets in this work, the scope of interest includes both $\mathcal{D}^r = \mathcal{C} \setminus \mathcal{D}^f$, the retention classes at the same level as the forgetting ones, and their parent classes in the taxonomy, represented as $\mathcal{D}^r_{\text{Parent}}$ and $\mathcal{D}^f_{\text{Parent}}$. Specifically, the utility of an unlearned model is $U = (\bar{A}(\mathcal{D}^r) + \bar{A}(\mathcal{D}^r_{\text{Parents}}) + \bar{A}(\mathcal{D}^f_{\text{Parents}}))/3$, where $\bar{A}$ is the same scaled accuracy function as used in defining the forgetting quality. We then define a Q-U score as the harmonic mean of quality and utility, inspired by the F-score: Q-U $= 2QU/(Q + U)$.

**Preservation of general capabilities.** Radford et al. (2021) demonstrated CLIP's remarkable zero-shot image classification performance over multiple datasets, which should not be impaired by the requested unlearning as long as those class labels have no overlap with the forgetting set $\mathcal{D}^f$. To test this general ability of unlearned CLIP, we follow (Radford et al., 2021; Khattak et al., 2023) to use several image classification datasets (Krizhevsky et al., 2009; Fei-Fei et al., 2004; Nilsback & Zisserman, 2008; Krause et al., 2013; Parkhi et al., 2012; Bossard et al., 2014) to assess the zero-shot classification performance of the model.

### 5.2 Results

**Main comparison results.** Table 2 shows the results of various MU baselines on the ImgnetDogs dataset. **GA-based** methods achieve high forgetting quality but suffer from a significant drop in retained fine-grained concept recognition accuracy due to their unbounded optimization loss. Without a regularization term, the fine-grained accuracy on the retention set drops sharply to 1.57 %. Introducing a KL-divergence regularization term on the forget set helps preserve utility, raising the retention set accuracy to 35.96 %. **Relabeling** performs poorly in fine-grained unlearning, exhibiting low forgetting quality and model utility. The Q-U

Table 3: Fine-grained concept removal results vs. different difficulty levels on ImgnetDogs.

| Difficulty Level | Method | $\mathcal{D}_{test}^{f}$ | | $\mathcal{D}_{test}^{r}$ | | Performance Metrics | | |
|---|---|---|---|---|---|---|---|---|
| | | coarse ↑ | fine ↓ | coarse ↑ | fine ↑ | Quality ↑ | Utility ↑ | Q-U ↑ |
| Difficult | CLIP (Radford et al., 2021) | 86.20 | 93.40 | 50.88 | 65.55 | – | – | – |
| | HGA+KL (Ours) | 88.20 | 2.00 | 48.23 | 54.56 | 97.86 | 92.68 | 95.20 |
| Medium | CLIP (Radford et al., 2021) | 75.00 | 82.80 | 52.13 | 66.74 | – | – | – |
| | HGA+KL (Ours) | 6.00 | 0.40 | 50.29 | 58.29 | 99.52 | 94.61 | 97.00 |
| Easy | CLIP (Radford et al., 2021) | 60.73 | 75.82 | 53.66 | 67.42 | – | – | – |
| | HGA+KL (Ours) | 64.36 | 0.73 | 52.21 | 63.09 | 99.04 | 96.95 | 97.98 |

Table 4: Coarse-grained concept removal results on ImgnetDogs.

| Method | $\mathcal{D}_{test}^{f}$ | | $\mathcal{D}_{test}^{r}$ | | Performance Metrics | | | |
|---|---|---|---|---|---|---|---|---|
| | coarse ↓ | fine ↓ | coarse ↑ | fine ↑ | Quality ↑ | Utility ↑ | Q-U ↑ | Zero-shot ↑ |
| Origin CLIP (Radford et al., 2021) | 76.75 | 80.25 | 53.45 | 67.32 | – | – | – | 83.24 |
| GA (Jang et al., 2022) | 0.00 | 0.00 | 10.33 | 29.07 | **100.00** | 31.25 | 47.62 | 80.52 |
| GDiff (Liu et al., 2022) | 0.00 | 0.00 | 10.66 | 28.46 | 100.00 | 31.11 | 47.46 | 81.20 |
| GA+KL (Yao et al., 2023) | 0.00 | 0.25 | 42.44 | 40.84 | 99.84 | 70.03 | 82.32 | 82.30 |
| Relabeling (Golatkar et al., 2020) | 14.75 | 16.00 | 41.65 | 43.64 | 80.42 | 71.37 | 75.63 | 81.02 |
| SalUn (Fan et al., 2023) | 7.25 | 29.50 | 51.69 | 59.30 | 76.90 | 92.40 | 83.94 | 82.55 |
| ME+GD (Yuan et al., 2024) | 22.50 | 31.25 | 50.18 | 50.15 | 65.87 | 84.19 | 73.91 | 82.19 |
| Task Vector (Ilharco et al., 2022) | 8.75 | 28.00 | 55.45 | 63.70 | 76.85 | **99.18** | 86.60 | 82.64 |
| NPO+KL (Zhang et al., 2024b) | 8.25 | 24.50 | 55.03 | 63.03 | 79.36 | 98.29 | 87.82 | **83.14** |
| HGA+KL (Ours) | 10.50 | 8.00 | 52.35 | 60.09 | 88.18 | 93.60 | **90.81** | 82.99 |

score of **SalUn** is better than the relabeling method (62.43 % vs. 56.30 %). The **ME** method disrupts the intrinsic relationships among fine-grained concepts, leading to a significant reduction in the accuracy of the retained concepts. The **task-vector** struggles to unlearn fine-grained concepts, resulting in low forgetting quality while maintaining high model utility. Unlike the unbounded loss in the GA-based method, the unlearning optimization loss for **NPO** is bounded, avoiding catastrophic collapse and achieving better unlearning performance. Our proposed method (**HGA**), incorporating KL divergence, attains a Q-U score of 95.20 %, nearly 3 % higher than the NPO method.

We also report the average zero-shot classification accuracy of the unlearned model. The results indicate that forgetting specific fine-grained concepts generally does not significantly impair the model's generalizability, except in the case of the GA method without regularization, which experiences notable degradation. Moreover, models employing relabeling-based unlearning methods exhibit a more pronounced decline in generalizability.

**Unlearning results for the fine-grained forgetting classes of various difficulty levels.** Like humans, FMs demonstrate varying degrees of memorization for concepts, leading to different difficulty levels for unlearning. In our case study, we quantify concept memorization using the model's confidence scores about the concepts, offering a simpler alternative to traditional metrics (Zhao et al., 2024; Zhao & Triantafillou, 2024). We conduct three sets of experiments under difficult, medium, and easy unlearning settings, corresponding to decreasing average confidence scores of the concepts to be unlearned. As shown in Table 3, removing difficult, high-confidence concepts causes a more substantial drop in model utility compared to easy, low-confidence ones. This highlights the importance of avoiding excessive unlearning of low-confidence concepts and carefully regulating the unlearning of high-confidence concepts to preserve utility in future work.

**Unlearning results for the coarse-grained classes.** We further evaluate coarse-grained classes unlearning (e.g., Retriever and Setter), where unlearning requires removing both the coarse-grained classes and all corresponding fine-grained classes. We observe that, when unlearning coarse-grained classes, balancing unlearning quality and model utility is more challenging than for fine-grained classes, as reflected by a lower Q-U metric for our proposed method, shown in Table 4. Importantly, the difficulty of unlearning is not

determined solely by concept granularity but also by the model's degree of memorization. For example, CLIP performs worse on coarse-grained categories than on fine-grained ones. As a result, for some methods, unlearning fine-grained concepts under a coarse-grained class can require greater effort than unlearning the coarse-grained concept itself, highlighting that unlearning difficulty depends jointly on concept granularity and the target model's idiosyncrasies.

**Results with varying numbers of forgetting training samples.** Table 5 illustrates the influence of varying the number of forgetting training samples on the unlearning performance of our proposed method.

When the number of forgetting training samples is too small—such as only 10 images per category—achieving effective unlearning is challenging, resulting in lower forget quality (70%). Unlearning quality improves as the number of forgotten samples increases; however, this comes at the cost of reduced model utility. Notably, the improvement in unlearning effectiveness becomes less significant beyond 30 samples, highlighting the sample efficiency of our proposed unlearning method.

Table 5: Unlearning performance with different numbers of forgotten training samples per fine-grained class.

| Samples Number | Quality ↑ | Utility ↑ | Q-U ↑ |
|---|---|---|---|
| 10 | 70.02 | 95.58 | 80.83 |
| 20 | 80.09 | 94.97 | 86.89 |
| 30 | 93.36 | 94.20 | 93.78 |
| 50 | 94.65 | 93.69 | 94.17 |
| 100 | 95.72 | 92.58 | 94.12 |
| 150 | 96.15 | 93.19 | 94.65 |
| 200 | 97.86 | 92.68 | 95.20 |

**Limitation of data-tracing MU methods.** While we applied data-tracing MU methods to the case study, we contend they exhibit significant limitations for the knowledge-tracing FMU. Most existing data-tracing MU methods yield a subpar quality-utility trade-off and zero-shot generalization in our case study. Although NPO and our proposed method perform better than others in the quality-utility trade-off, they

Table 6: Comparison of unlearning performance (Q-U metrics) on the OOD dataset.

| Methods | Quality ↑ | Utility ↑ | Q-U ↑ |
|---|---|---|---|
| GDiff | 87.57 | 77.02 | 81.96 |
| GA+KL | 85.24 | 84.50 | 84.87 |
| NPO+KL | 34.98 | 96.60 | 51.37 |
| HGA+KL (Ours) | 27.20 | 97.92 | 42.58 |

have poor robustness under the out-of-dataset test (Table 6), where models were unlearned on ImgnetDogs and evaluated on OxfordPet. The results show that all data-tracing MU methods, including ours, fail to tackle knowledge-tracing MU, which underscores the limitations of current data-tracing MU methods. We expect that future techniques will be natively designed for knowledge-tracing FMU.

### 5.3 Discussions

Our case study uses a taxonomy to represent knowledge structures for its flexibility in lieu of its completeness. We can extend it to higher abstraction levels, such as forgetting *retriever* while retaining *dog.* One can also refine it further by subdividing *golden retriever* into finer-grained categories or attributes. In this structure, each abstract concept corresponds to an inner node, and the granularity of the hierarchy determines the specificity of knowledge encoded in the leaf nodes. We acknowledge that a real-world ontology should be more complex than ours. A knowledge graph embedded in an LLM can be exponentially large. Exploring alternative structural representations and unlearning setups, such as graph-based knowledge unlearning, is a promising direction for future research on the knowledge-tracing FMU.

Finally, we present additional knowledge-tracing unlearning scenarios and some potential strategies for constructing corresponding forgetting and retention datasets as follows. **Retrieval:** Forgetting targets are visual concepts such as "Golden Retriever." The forgetting dataset consists of image-text pairs related to the target concepts, with images sourced from public datasets and captions generated by proprietary VLMs (OpenAI, 2024) and verified by humans. The retention dataset includes semantically similar but distinct concepts (e.g., other dog breeds) to assess the specificity of forgetting. General vision-language benchmarks (Chen et al., 2015) can be used to evaluate overall generalization. **VQA:** (e.g., LLaVA (Liu et al., 2023)). Forgetting targets include visual entities such as "Donald Trump." The forgetting dataset comprises images of the target paired with QA examples—open-ended or multiple-choice—generated using GPT-4o (OpenAI, 2024) and verified manually. The retention dataset involves QA pairs about related but different concepts (e.g., other public figures). General VQA benchmarks (Fu et al., 2023) assess broader

reasoning abilities. **Text QA:** (e.g., LLaMA (Touvron et al., 2023)). Forgetting targets are private entity-level facts, such as details about "Harry Potter" characters. The forgetting dataset consists of QA pairs or passages explicitly referencing those facts, generated or collected to ensure contextual diversity. The retention set includes text about similar but untargeted entities. Evaluation relies on QA datasets such as Natural Questions (Kwiatkowski et al., 2019) and TriviaQA (Joshi et al., 2017). We leave the specific implementation and study of these cases to future work.

# 6 Alternative Views

While we argue to prioritize the research on knowledge-tracing FMU, one might argue that the data-tracing MU should remain the top priority even for FMs because the resulting methods are generally applicable. Indeed, we anticipate that the unlearning methods in the proposed knowledge-tracing paradigm will still rely on data for unlearning. One might also have a different view about the insights we draw from cognitive science. Airplanes fly in a way different from how birds fly. Hence, it is not necessary to design FMU frameworks following the human brain's forgetting mechanism.

There could also be a wild alternative view that FMs do not need unlearning because the scaling law and hardware innovation allow them to continually grow and learn new information without losing previously acquired capabilities. Instead of prioritizing research on FMU, the focus should be on continual learning of FMs, where selective forgetting could be a subtopic or a natural property emerging in an FM's continual learning process.

Another research priority one would probably like to pursue is evaluation at MU. We have witnessed some works on this topic already (Thaker et al., 2024; Thudi et al., 2022b; Shi et al., 2024), which call for more comprehensive and solid benchmarks for MU research. In the data-tracing MU, one can obtain an oracle model by retraining a model over the retention set. However, such a model is often not supplied with any existing MU benchmarks, and it remains unclear how to leverage the oracle model to evaluate MU methods. Currently, there is no widely accepted standard for evaluating knowledge-level unlearning. Through this position paper, we hope to inspire future work that advances the evaluation criteria.

# 7 More related work

Besides the works reviewed in Section 2, our position and case study are also related to the following works.

**MU on vision.** The SISA framework (Bourtoule et al., 2021) has advanced MU in the classification task, with subsequent efforts (Wu et al., 2020; Yan et al., 2022) enhancing retraining efficiency. Recent research has shifted towards approximate MU that modifies trained models directly. Early approaches employing Hessian approximations (Guo et al., 2019; Sekhari et al., 2021) faced high computation costs. More general methods have been introduced for class-wise unlearning in deep neural networks (Chen et al., 2023; Lin et al., 2023; Kurmanji et al., 2024; Fan et al., 2023; Liu et al., 2024a). The concept of MU has also been extended to diffusion models (Gandikota et al., 2023; Park et al., 2024; Gong et al., 2025; Zhang et al., 2024c), aiming to prevent generating harmful or unethical content.

**MU for LLMs.** How to remove the influence of undesirable data on the pre-trained LLMs (Liu et al., 2024b; Shi et al., 2024; Huu-Tien et al., 2024; Li et al., 2024d; Jin et al., 2024; Qiu et al., 2024; Cha et al., 2025) has received growing attention. Various unlearning methods have been proposed, including gradient ascent (Jang et al., 2022), random relabeling (Yao et al., 2024; 2023), and regenerating desirable answers (Eldan & Russinovich, 2023) or safe tokens (Ishibashi & Shimodaira, 2023), demonstrating effective unlearning capabilities. Additionally, approaches combining gradient ascent with KL divergence (Wang et al., 2023; Chen & Yang, 2023; Yao et al., 2024) or gradient descent (Yao et al., 2024; Chen & Yang, 2023) have been widely adopted. Task-vector-based techniques (Zhang et al., 2023; Liu et al., 2024c; Hu et al., 2024) and weight-importance strategies (Wu et al., 2023; Yu et al., 2023) further enhance unlearning precision while preserving utility. Input-based unlearning methods (Pawelczyk et al., 2023; Huang et al., 2024c) have emerged as a complementary solution for black-box LLMs unlearning.

**Multi-modality MU.** Compared to single modality MU, unlearning for multimodal vision-language models (Cheng & Amiri, 2025; Li et al., 2024c; Ma et al., 2024; Yang et al., 2024; Poppi et al., 2025) remains largely underexplored. SIU (Li et al., 2024c) proposed an efficient method for unlearning visual concepts in the pre-trained LLaVA (Liu et al., 2023) using just one image during the training process. MMDelete (Cheng & Amiri, 2025) proposed a multi-modality unlearning method for fine-tuned FMs on image-text and graph-text datasets. CLIPErase (Yang et al., 2024) and Safe-CLIP (Poppi et al., 2025) explored machine unlearning on the CLIP model. Inspired by TOFU (Maini et al., 2024), a new benchmark FIUBENCH (Ma et al., 2024), which contains fictitious facial identity data, has been proposed to evaluate the unlearning methods on the fine-tuned VLM.

**Model editing.** Model editing, or knowledge editing (Mitchell et al., 2022; Huang et al., 2024b; Wang et al., 2024c), shares similarities with unlearning, as both seek to modify the model while preserving its generalization capabilities. However, the two processes differ fundamentally: model editing focuses on predefined updates to address hallucinations in pre-trained models, whereas unlearning involves removing information without predefined outputs. While much of the existing research has concentrated on editing large language models (Mitchell et al., 2021; 2022; Wang et al., 2024c), recent efforts have introduced new benchmarks for editing VLMs (Huang et al., 2024b; Zhang et al., 2024a; Huang et al., 2024a; Li et al., 2024b).

## 8    Conclusion

This position paper is on the confluence of MU and FMs, or FMU in short. We have provided a historical review of MU and FMU, which exposes that existing works trace data — removing specific training examples' influence from FMs. We argue that this setup is impractical for many FM users because they have no or limited access to FMs' massive training data. Instead, we advocate for a shift toward knowledge-tracing FMU to meet diverse unlearning requests from all FM stakeholders. Besides this argument from a practical view, we also draw insights from cognitive science, backing that knowledge-tracing FMU aligns with human-like memory processes. We further discuss the nontrivial challenges inherent in the knowledge-tracing machine unlearning paradigm. We have provided a detailed case study about CLIP, a visual-language FM, to explore our position further. The unlearning requests are formalized about the removal of some specific fine-grained object class recognition capabilities. We encourage the research community to pay attention to what to unlearn (knowledge or data) when they expand investigations into MU and FMU.

**Acknowledgments:**   We sincerely thank the reviewers and editor(s) for their insightful feedback on improving the work, which was partially supported by a Gemini Academic Program Award and Azure Sponsorship.

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

# A    Appendix

## A.1    Further Details of Related Work

In this section, we provide more details on the unlearning setups of existing unlearning work. We systematically categorize unlearning tasks, models, and targets of related papers in Table 7.

Table 7: Experiment setup details for existing machine unlearning work.

| Related Work | Task | Unlearned Model/Target |
|---|---|---|
| Golatkar et al. (2020) | Image classification | All-CNN/Entire Class or a hundred images of the class |
| Jang et al. (2022) | Unlearn Privacy Information | GPT-Neo/Privacy Instances |
| Chen et al. (2023) | Image classification | All-CNN and Resnet/Entire Class |
| Lin et al. (2023) | Image classification | Resnet/Entire Class |
| Fan et al. (2023) | Image classification and generation | Resnet and DDPM/Random samples and Entire class |
| Chen & Yang (2023) | Classification and Summarization | Fine-tuned T5 and T3 model/Random Instances |
| Zhang et al. (2023) | Reduce the toxicity | GPT-2 Model/All instances |
| Gandikota et al. (2023) | Text-to-Image generation | Stable Diffusion Model/Predefined Concepts |
| Wang et al. (2024b) | Synthetic author profiles QA | Fine-tuned Llama-2-7B/Random Entities |
| Maini et al. (2024) | Synthetic author profiles QA | Fine-tuned Llama-2-7B/Random Entities |
| Zhang et al. (2024b) | Synthetic author profiles QA | Fine-tuned Llama-2-7B/Random Entities |
| Wu et al. (2023) | Privacy information forgetting | Fine-tuned BERT-base model/All instances |
| Eldan & Russinovich (2023) | Unlearn the Harry Potter books | Pre-trained Llama2-7b model/All instances |
| Yao et al. (2023) | Unlearn the Harry Potter books | Fine-tuned Llama model/All instances |
| Yao et al. (2024) | Removing copyrighted data | Pre-trained Yi-6B /Pre-training samples |
| Jin et al. (2024) | Remove celebrity information | Pre-trained LLaMA3 and Phi-3/Predefined entities |
| Li et al. (2024c) | Unlearn visual concepts | Pre-trained LLaVA/Predefined visual concepts |
| Li et al. (2024d) | Remove hazardous knowledge | Pre-trained ZEPHYR-7B and YI-34B/Hazardous VQA |
| Poppi et al. (2025) | Unlearn unsafe embeddings | Pre-trained CLIP/Unsafe Images and Texts |

## A.2    More details of the Dataset

Table 8:   Hierarchy Fine-grained Recognition Dataset Details

| Dataset | Coarse Num. | Fine Num. | Training Num. | Testing Num. |
|---|---|---|---|---|
| CompCars-S | 48 | 292 | 26,630 | 8,943 |
| ImgnetDogs | 14 | 99 | 19,800 | 4,950 |

**CompCars-S.** The original dataset comprises 161 coarse and 1687 fine classes; however, the classification accuracy across these classes is notably low. Some coarse-grained categories may contain only one fine-grained category, and some fine-grained categories have limited images. Consequently, we implemented a filtering process on the original dataset. The process is as follows: Initially, at the coarse-grained level, each category must include at least two fine-grained categories, and each fine-grained category must contain no fewer than 90 images; otherwise, the category would be removed. Subsequently, we utilized a pre-trained CLIP model (ViT-L/14) to refine the dataset further. Those images and car models are retained if the accuracy of the fine class is above 20%. Otherwise, the corresponding car model categories and images are removed. The details of dataset information are presented in Table 8.

**ImgnetDogs.** The construction of the ImgnetDogs dataset is based on WordNet (Fellbaum, 1998). The StanfordDogs dataset, as introduced in (Khosla et al., 2011), is also a fine-grained dog breed recognition

dataset, which forms a subset of ImageNet. However, some fine-grained dog categories in the StanfordDogs datasets are assigned to highly abstract coarse categories across different semantic levels. We selectively chose fine-grained categories with clear, well-defined, higher-level coarse semantic information from the original ImageNet dataset.

### A.3   More Details of the Case Study Setting

Unlearning fine-grained concepts that the model initially fails to recognize or has low accuracy is meaningless. Therefore, the selected concepts for unlearning should meet a predefined accuracy threshold. In our case study, we focus on unlearning fine-grained classes with an accuracy above 90%. For the medium and easy unlearning settings in ImgnetDogs, the overall accuracy of the unlearned fine-grained classes is 82% and 75%, respectively. The specific fine-grained concepts unlearned for each dataset are detailed in Table 9.

Table 9: Unlearned fine-grained concepts for each dataset.

| Dataset | Unlearn Fine Classes |
|---------|----------------------|
| CompCars-S | Acura MDX, Lexus RX, Jaguar XK, MINI CABRIO, Audi A7, Audi A5 coupe, Cadillac SRX, Corvette, Mustang |
| ImgnetDogs Difficult | German short-haired pointer, Boston terrier, West Highland white terrier, Labrador retriever, golden retriever, German shepherd dog, keeshond, Samoyed, Pomeranian, Border terrier |
| ImgnetDogs Medium | Irish setter, Gordon setter, basset hound, Airedale terrier, Shih-Tzu, miniature pinscher, Alaskan malamute, flat-coated retriever, Chesapeake Bay retriever, Sealyham terrier |
| ImgnetDogs Easy | English setter, beagle, whippet, Ibizan hound, Dandie Dinmont terrier, standard poodle, Border collie, Blenheim spaniel, cairn terrier, Doberman, groenendael |

### A.4   Baseline Machine Unlearning Methods

**Gradient Ascent.** Gradient Ascent (Jang et al., 2022; Thudi et al., 2022a; Kurmanji et al., 2024) is a straightforward yet effective unlearning method applied to various unlearning settings. GA aims at maximizing the predicted loss on the forgetting set, which can be formulated as follows:

$$\mathcal{L}_{GA} = \sum_{(x_i, y_i) \in \mathcal{D}^f} [log(y_i | x_i, \theta)]. \tag{4}$$

**Gradient Difference.** Gradient Difference (Liu et al., 2022) introduces the regularization term on the retaining dataset, which helps maintain the model ability on the retaining dataset. By incorporating the GA loss alongside the GD loss, the GDiff objective can be formulated as:

$$\mathcal{L}_{GD} = \sum_{(x_i, y_i^c) \in \mathcal{D}^f} \left[ - \log(y_i^c | x_i, \theta) \right], \tag{5}$$

$$\mathcal{L}_{GDiff} = \mathcal{L}_{GA} + \mathcal{L}_{GD}. \tag{6}$$

**KL Minimization.** Different from GD, KL minimization (Yao et al., 2023) minimizes the KL divergence between the prediction of the unlearned model and the origin model on the retaining dataset. The objective is defined as:

$$\mathcal{L}_{KL} = \sum_{(x_i, y_i^c) \in \mathcal{D}^f} \text{KL}(p_{\theta_0}(y_i^c | x_i) || p_\theta(y_i^c | x_i)). \tag{7}$$

**Random Labeling.** By fine-tuning the original model using relabeled labels (Golatkar et al., 2020) on the forgetting dataset, the relabeling method overwrites the information associated with the original labels. The optimization objective for relabeling is as follows:

$$\mathcal{L}_{Relabel} = \sum_{(x_i,.) \in \mathcal{D}^f} [-log(y^{rand}|x_i, \theta)], \tag{8}$$

where $y^{rand}$ is randomly chosen from the label set and $y^{rand} \neq y^f$.

**Negative Preference Optimization.** To address the catastrophic collapse problem of GA, NPO (Zhang et al., 2024b) has introduced the bounded optimization loss defined as

$$\mathcal{L}_{NPO} = -\frac{2}{\beta} \sum_{(x_i,y_i) \in \mathcal{D}^f} [log\sigma(-\beta log \frac{p_\theta(y_i|x_i)}{p_{\theta_0}(y_i|x_i)}]. \tag{9}$$

**Task Vectors.** Task vector (Ilharco et al., 2022; Liu et al., 2024c) first computes the forgetting set-specific vector defined as

$$\tau^f = \theta_{tune} - \theta_0, \tag{10}$$

where $\theta_{tune}$ stands for the model tuned on the forgetting set $\mathcal{D}^f$ and $\theta_0$ represent the origin trained model. Subsequently, the task vector is negated and applied to the original model weights to compute the unlearned model as follows

$$\theta^u = \theta_0 - \alpha\tau^f. \tag{11}$$

**SalUn.** SalUn (Fan et al., 2023) introduces a gradient-based weight saliency map to identify important parameters for unlearning. The saliency map is defined as:

$$m_s = \mathbb{I}[\nabla_\theta \mathcal{L}(\theta, \mathcal{D}^f)_{\theta=\theta_0} > \alpha], \tag{12}$$

where $\mathbb{I}[\cdot]$ denotes the indicator function and $\alpha$ is a predefined threshold controlling the selection. The method selectively updates parameters with high gradient magnitudes using a relabeling strategy while freezing the remaining parameters to preserve the model's utility.

**ME.** ME (Yuan et al., 2024) minimizes the output distribution of the unlearned model between the uniform distribution, which is defined as:

$$\mathcal{L}_{ME} = \sum_{(x_i,y_i) \in \mathcal{D}^f} \text{KL}(\mathcal{U}_K||p_\theta(y_i|x_i)) \tag{13}$$

where $U_K$ is the uniform distribution over the classes.

### A.5   Training Details

We use a pre-trained ViT-L/14 CLIP model as the base model in all experiments. The prompts for each dataset are provided in Table 10. The unlearning process is trained for 8 epochs using the Adam optimizer. The batch size is set to 32 for the ImgnetDogs dataset and 16 for CompCars-S. For GA-based methods, the initial learning rate (lr) is set to $8 \times 10^{-8}$, for SalUn, it is $2 \times 10^{-7}$, and for all other methods, it is $1 \times 10^{-7}$. We save the checkpoint for evaluation when the unlearning accuracy on the training set stops decreasing. All experiments are conducted on a single Nvidia RTX A6000 GPU. Additional training details for the baseline methods are provided in Table 11. Since no retain set is used during training, KL divergence and gradient ascent are applied solely to the forget set to preserve the model's coarse recognition capabilities.

### A.6   More results

### A.6.1   More results on the ImgnetDogs dataset.

Details of zero-shot classification results are shown in Table 12. We evaluated several unlearning methods on the OxfordPet dataset, regarded as an out-of-domain evaluation dataset. According to the results shown

Table 10: Prompts of CompCars-S and ImgnetDogs dataset.

| Dataset | Prompts |
|---|---|
| CompCars-S | 'a photo of a {}', 'a photo of the {}', 'a photo of my {}', |
| | 'i love my {}!', 'a photo of my dirty {}', 'a photo of my clean {}', |
| | 'a photo of my new {}', 'a photo of my old {}' |
| ImgnetDogs | 'a bad photo of a {}', 'a photo of many {}', 'a sculpture of a {}', |
| | 'a photo of the hard to see {}', 'a low resolution photo of the {}', 'a rendering of a {}', |
| | 'graffiti of a {}', 'a bad photo of the {}', 'a cropped photo of the {}', |
| | 'a tattoo of a {}', 'the embroidered {}', 'a photo of a hard to see {}', |
| | 'a bright photo of a {}', 'a photo of a clean {}', 'a photo of a dirty {}', |
| | 'a dark photo of the {}', 'a drawing of a {}', 'a photo of my {}', |
| | 'the plastic {}', 'a photo of the cool {}', 'a close-up photo of a {}', |
| | 'a black and white photo of the {}', 'a painting of the {}', 'a painting of a {}', |
| | 'a pixelated photo of the {}', 'a sculpture of the {}', 'a bright photo of the {}', |
| | 'a cropped photo of a {}', 'a plastic {}', 'a photo of the dirty {}', |
| | 'a jpeg corrupted photo of a {}', 'a blurry photo of the {}', 'a photo of the {}', |
| | 'a good photo of the {}', 'a rendering of the {}', 'a {} in a video game', |
| | 'a photo of one {}', 'a doodle of a {}', 'a close-up photo of the {}', |
| | 'a photo of a {}', 'the origami {}', 'the {} in a video game', |
| | 'a sketch of a {}', 'a doodle of the {}', 'an origami {}', |
| | 'a low resolution photo of a {}', 'the toy {}', 'a rendition of the {}', |
| | 'a photo of the clean {}', 'a photo of a large {}', 'a rendition of a {}', |
| | 'a photo of a nice {}', 'a photo of a weird {}', 'a blurry photo of a {}', |
| | 'a cartoon {}', 'art of a {}', 'a sketch of the {}', |
| | 'an embroidered {}', 'a pixelated photo of a {}', 'itap of the {}', |
| | 'a jpeg corrupted photo of the {}', 'a good photo of a {}', 'a plushie {}', |
| | 'a photo of the nice {}', 'a photo of the small {}', 'a photo of the weird {}', |
| | 'the cartoon {}', 'art of the {}', 'a drawing of the {}', |
| | 'a photo of the large {}', 'a black and white photo of a {}', 'the plushie {}', |
| | 'a dark photo of a {}', 'itap of a {}', 'graffiti of the {}', |
| | 'a toy {}', 'itap of my {}', 'a photo of a cool {}', |
| | 'a photo of a small {}', 'a tattoo of the {}' |

Table 11: Training details and hyper-parameters of the baselines

| Method | Optimization Loss function | Lr | Hyper Parameters |
|---|---|---|---|
| GA | $\mathcal{L}_{GA}(x^f, y^f)$ | 8e-8 | - |
| GDiff | $\mathcal{L}_{GA}(x^f, y^f) + \mathcal{L}_{GD}(x^f, y^f_c)$ | 8e-8 | - |
| ME+GD | $\mathcal{L}_{ME}(x^f, y^f) + \mathcal{L}_{GD}(x^f, y^f_c)$ | 1e-7 | - |
| Task Vector | $\mathcal{L}_{GD}(x^f, y^f) + 0.05 * \mathcal{L}_{GA}(x^f, y^f_c)$ | 1e-7 | $\alpha = 1.5$ |
| KL | $\mathcal{L}_{GA}(x^f, y^f) + \alpha_c \text{KL}(x^f, y^f_c) + \alpha_f \text{KL}(x^f, y^f)$ | 8e-8 | $\alpha_c = 5, \alpha_f = 20$ |
| NPO+KL | $\mathcal{L}_{NPO}(x^f, y^f) + \alpha_c \text{KL}(x^f, y^f_c) + \alpha_f \text{KL}(x^f, y^f)$ | 1e-7 | $\beta = 0.5, \alpha_c = 5, \alpha_f = 20$ |
| HGA+KL | $\mathcal{L}_{HGA}(x^f, y^f) + \alpha_c \text{KL}(x^f, y^f_c) + \alpha_f \text{KL}(x^f, y^f)$ | 1e-7 | $m = 2, \alpha_c = 10, \alpha_f = 20$ |
| Relabel | $\mathcal{L}_{Relabel}(x^f, .)$ | 1e-7 | - |
| SalUn | $\mathcal{L}_{Relabel}(x^f, .)$ | 2e-7 | $\alpha = 0.1$ |

in Table 13, nearly all unlearning methods struggled to achieve high-quality forgetting, except for GA-based methods. While GA-based methods demonstrated superior unlearning performance, they significantly decreased performance on non-unlearned fine-grained concepts. Since the CLIP model is a non-generative model, its classification evaluations are based on a closed set, requiring predefined class names for testing. The limited number of categories in the OxfordPet dataset compared to the training set also impacts the performance of these unlearning methods. Future work will improve the unlearning method further and expand this case study to generative models (Li et al., 2024a; Wang et al., 2024a) with fine-grained recognition capabilities.

Table 12: Generalization performance across different baseline methods for the unlearned model.

| Dataset | Stanford Cars | Food101 | Flower102 | Caltech101 | Cifar100 | Avg ↑ |
|---|---|---|---|---|---|---|
| Origin CLIP (Radford et al., 2021) | 77.75 | 92.32 | 79.18 | 91.11 | 75.82 | 83.24 |
| GA (Jang et al., 2022) | 75.43 | 89.26 | 74.42 | 89.73 | 63.90 | 78.55 |
| GDiff (Liu et al., 2022) | 77.10 | 90.75 | 77.36 | 90.57 | 68.67 | 80.89 |
| GA+KL(Yao et al., 2023) | 76.59 | 91.47 | 78.08 | 90.78 | 71.40 | 81.66 |
| NPO+KL (Zhang et al., 2024b) | 77.07 | 91.90 | 78.26 | 90.65 | 73.12 | 82.20 |
| Relabeling (Golatkar et al., 2020) | 76.88 | 91.38 | 76.26 | 89.25 | 72.81 | 81.32 |
| Task Vector (Ilharco et al., 2022) | 77.15 | 92.05 | 78.53 | 90.28 | 74.85 | **82.57** |
| SalUn (Fan et al., 2023) | 77.50 | 91.56 | 76.66 | 89.31 | 73.83 | 81.77 |
| ME+GD (Yuan et al., 2024) | 77.14 | 91.56 | 76.48 | 89.50 | 73.80 | 81.70 |
| HGA+KL (Ours) | 77.24 | 92.00 | 78.81 | 90.68 | 73.94 | 82.53 |

Table 13: Comparison of fine-grained concept removal results across different baseline methods on the OOD dataset.

| Setting | $\mathcal{D}^f_{test}$ | | $\mathcal{D}^r_{test}$ | | Performance Metrics | | |
|---|---|---|---|---|---|---|---|
| | coarse ↑ | fine ↓ | coarse ↑ | fine ↑ | Quality ↑ | Utility ↑ | Q-U ↑ |
| Origin CLIP (Radford et al., 2021) | 92.18 | 99.10 | 73.98 | 91.54 | – | – | – |
| GDiff (Liu et al., 2022) | 85.77 | 12.32 | 54.77 | 58.59 | 87.57 | 77.02 | 81.96 |
| GA+KL (Yao et al., 2023) | 87.78 | 14.63 | 63.31 | 66.57 | 85.24 | 84.50 | 84.87 |
| NPO+KL (Zhang et al., 2024b) | 94.09 | 64.43 | 69.34 | 87.94 | 34.98 | 96.60 | 51.37 |
| HGA+KL (Ours) | 93.59 | 72.14 | 72.15 | 88.09 | 27.20 | 97.92 | 42.58 |

Additionally, we provide additional results for the medium and easy unlearning settings, as shown in Table 14 and Table 15. Across different memorization settings, our method consistently performs the best. Additionally, the relabeling-based methods consistently show the poorest performance. The task-vector method performs well in both medium and easy settings, indicating that it is unsuitable for high-memorization concept unlearning. Furthermore, the NPO method's forgetting quality is not very high in low memorization settings, demonstrating its limitation.

### A.6.2 More results on CompCars-S dataset

The comparison results of different baseline methods on the CompCars-S dataset are presented in Table 16 and Table 17. In this dataset, gradient ascent outperforms the KL divergence method. Additionally, relabeling-based methods fail to achieve effective unlearning, similar to their performance on the ImgnetDogs dataset. Notably, our proposed method significantly outperforms other unlearning techniques on the CompCars-S dataset. Moreover, the generalizability of most unlearned models remains largely unaffected, except for the relabeling-based method and the gradient ascent method without regularization, both of which exhibit substantial degradation.

Table 14: Comparison of fine-grained concept removal results across different baseline methods on ImgnetDogs (Medium Unlearn).

| Method | $\mathcal{D}^f_{test}$ | | $\mathcal{D}^r_{test}$ | | Performance Metrics | | |
|---|---|---|---|---|---|---|---|
| | coarse ↑ | fine ↓ | coarse ↑ | fine ↑ | Quality ↑ | Utility↑ | Q-U ↑ |
| Origin CLIP (Radford et al., 2021) | 75.00 | 82.80 | 52.13 | 66.74 | – | – | – |
| GA (Jang et al., 2022) | 22.2 | 0.00 | 30.70 | 2.63 | 100.00 | 30.81 | 47.11 |
| GDiff (Liu et al., 2022) | 58.4 | 0.00 | 41.05 | 18.05 | 100.00 | 61.22 | 75.95 |
| GA+KL (Yao et al., 2023) | 69.00 | 1.20 | 49.01 | 40.38 | 98.55 | 82.18 | 89.62 |
| NPO+KL (Zhang et al., 2024b) | 74.6 | 4.40 | 50.20 | 57.30 | 94.69 | 93.88 | 94.28 |
| Relabeling (Golatkar et al., 2020) | 50.60 | 49.40 | 39.87 | 51.28 | 40.34 | 73.59 | 52.11 |
| Task vector(Ilharco et al., 2022) | 77.60 | 13.80 | 54.40 | 60.09 | 83.33 | 96.68 | 89.51 |
| SalUn(Fan et al., 2023) | 55.00 | 41.40 | 42.45 | 54.29 | 50.00 | 78.71 | 61.15 |
| ME+GD (Yuan et al., 2024) | 83.60 | 44.80 | 43.30 | 48.67 | 45.89 | 85.33 | 59.68 |
| HGA+KL (Ours) | 76.00 | 0.40 | 50.29 | 58.29 | 99.52 | 94.60 | 97.00 |

Table 15: Comparison of fine-grained concept removal results across different baseline methods on ImgnetDogs (Easy Unlearn).

| Method | $\mathcal{D}^f_{test}$ | | $\mathcal{D}^r_{test}$ | | Performance Metrics | | |
|---|---|---|---|---|---|---|---|
| | coarse ↑ | fine ↓ | coarse ↑ | fine ↑ | Quality ↑ | Utility ↑ | Q-U ↑ |
| Origin CLIP (Radford et al., 2021) | 60.73 | 75.82 | 53.66 | 67.42 | – | – | – |
| GA (Jang et al., 2022) | 24.55 | 0.00 | 18.39 | 3.89 | 100.00 | 26.82 | 42.29 |
| GDiff (Liu et al., 2022) | 71.09 | 0.00 | 45.41 | 6.73 | 100.00 | 64.87 | 78.69 |
| GA+KL (Yao et al., 2023) | 63.82 | 0.36 | 49.5 | 26.23 | 99.52 | 77.05 | 86.85 |
| NPO+KL (Zhang et al., 2024b) | 64.55 | 6.55 | 54.02 | 60.66 | 91.38 | 96.65 | 93.94 |
| Relabeling (Golatkar et al., 2020) | 37.09 | 32.18 | 33.18 | 44.57 | 57.55 | 63.00 | 60.16 |
| Task vector(Ilharco et al., 2022) | 68.36 | 4.91 | 48.59 | 60.86 | 80.10 | 97.94 | 88.12 |
| SalUn(Fan et al., 2023) | 39.64 | 28.19 | 34.98 | 45.55 | 62.82 | 66.00 | 64.37 |
| ME+GD (Yuan et al., 2024) | 86.18 | 42.18 | 53.80 | 49.18 | 44.36 | 90.98 | 59.64 |
| HGA+KL (Ours) | 64.36 | 0.73 | 52.21 | 63.09 | 99.04 | 96.95 | 97.98 |

Table 16: Comparison of fine-grained concept removal results across different baseline methods on CompCars-S.

| Method | $\mathcal{D}^f_{test}$ | | $\mathcal{D}^r_{test}$ | | Performance Metrics | | |
|---|---|---|---|---|---|---|---|
| | coarse ↑ | fine ↓ | coarse ↑ | fine ↑ | Quality↑ | Utility ↑ | Q-U ↑ |
| Origin CLIP (Radford et al., 2021) | 92.78 | 92.10 | 73.29 | 71.04 | – | – | – |
| GA (Jang et al., 2022) | 0.00 | 0.00 | 3.27 | 1.28 | 100.00 | 2.09 | 4.09 |
| GDiff (Liu et al., 2022) | 88.66 | 2.75 | 69.75 | 18.62 | 97.02 | 72.31 | 82.86 |
| GA+KL(Yao et al., 2023) | 45.02 | 1.38 | 41.93 | 8.21 | 98.51 | 39.10 | 55.98 |
| NPO+KL (Zhang et al., 2024b) | 89.69 | 16.15 | 70.13 | 39.82 | 82.46 | 82.80 | 82.63 |
| Relabeling (Golatkar et al., 2020) | 59.11 | 25.43 | 58.70 | 43.22 | 72.39 | 68.21 | 70.24 |
| Task vector(Ilharco et al., 2022) | 82.82 | 28.52 | 68.48 | 60.91 | 69.03 | 89.48 | 77.94 |
| SalUn(Fan et al., 2023) | 64.26 | 23.71 | 57.69 | 43.37 | 74.25 | 69.67 | 71.89 |
| ME+GD (Yuan et al., 2024) | 99.66 | 28.18 | 77.83 | 37.96 | 69.40 | 84.47 | 76.20 |
| HGA+KL (Ours) | 87.97 | 2.41 | 68.68 | 59.04 | 97.39 | 90.54 | **93.84** |

Table 17: Generalization performance across different baseline methods for the unlearned model.

| Dataset | Food101 | Flower102 | Caltech101 | OxfordPet | Cifar100 | Avg ↑ |
|---|---|---|---|---|---|---|
| Origin CLIP (Radford et al., 2021) | 92.32 | 79.18 | 91.11 | 93.59 | 75.82 | 86.40 |
| GA (Jang et al., 2022) | 92.19 | 78.71 | 90.92 | 93.57 | 73.18 | 85.71 |
| GDiff (Liu et al., 2022) | 92.29 | 79.61 | 91.01 | 93.76 | 74.32 | 86.20 |
| GA+KL(Yao et al., 2023) | 92.32 | 79.17 | 91.05 | 93.62 | 74.07 | 86.05 |
| NPO+KL (Zhang et al., 2024b) | 92.26 | 78.91 | 90.95 | 93.10 | 75.61 | **86.34** |
| Relabeling (Golatkar et al., 2020) | 91.77 | 76.99 | 90.18 | 90.11 | 73.17 | 84.44 |
| Task Vector (Ilharco et al., 2022) | 92.30 | 78.74 | 91.02 | 93.16 | 75.45 | 86.13 |
| SalUn(Fan et al., 2023) | 91.52 | 76.35 | 90.07 | 88.14 | 73.51 | 83.92 |
| ME+GD (Yuan et al., 2024) | 91.22 | 75.05 | 90.28 | 86.26 | 73.21 | 83.20 |
| HGA+KL (Ours) | 92.26 | 78.91 | 90.95 | 93.10 | 75.61 | 86.17 |

