# OpenReview forum: "Lifting Data-Tracing Machine Unlearning to Knowledge-Tracing for Foundation Models"
_TMLR — Accepted by TMLR_

### Review · Reviewer_SrE6 · 2025-11-01

**Summary Of Contributions:**

The paper proposes to extend machine unlearning (MU) from the traditional data-tracing formulation to a knowledge-tracing paradigm, aiming to forget higher-level knowledge or capabilities in foundation models (FMs). The authors motivate this proposal by arguing that FM stakeholders typically lack access to raw training data and thus can only specify unlearning requests in terms of concepts or knowledge. They support their position with a case study on CLIP, treating the forgetting of fine-grained categories while retaining coarse-grained categories as an instance of knowledge-level unlearning. Methodologically, the paper modifies standard data-tracing MU approaches by introducing a hinge loss and a KL regularization.

Strengths:
1. The topic of controllability and selective knowledge-level unlearning in foundation models is timely and of broad conceptual interest.
2. The paper is clearly written and well contextualized.
3. The empirical setup is reproducible.

Weaknesses:
1. The concept of knowledge-tracing unlearning is not formally defined. In the case study, authors operationalize “knowledge” merely as class categories, treating the removal of fine-grained classes as an instance of knowledge-level unlearning. While this offers a concrete proxy, it does not constitute a formal definition of “knowledge”. The notion remains underspecified without representation space, trace mechanism, or criterion for knowledge removal provided. As a result, the proposed “knowledge-tracing” setup effectively reduces to class-level data unlearning.
2. The implementation contradicts the main motivation: the proposed method still requires constructing D^f = {(x, y, y^c)}, implying full access to class-level data, which the introduction explicitly states should be unavailable.
3. The only methodological change seems to be a hinge loss plus KL regularization, which is a stabilization trick rather than a new paradigm.
4. The authors ultimately acknowledge that knowledge-tracing MU “will still rely on data” (Sec.5), undermining the conceptual distinction they initially promote.
5. This work remains a position paper with limited technical novelty.

**Audience:**

Yes

**Audience Explanation:**

TMLR readers interested in machine unlearning or controllable foundation models will likely find the discussion thought-provoking. The conceptual framing that recasts unlearning as a knowledge-level operation may inspire future research.

**Broader Impact Concerns:**

No major ethical concerns are identified. The work aims to enhance model privacy, which is a positive direction.

**Claims And Evidence:**

No

**Claims Explanation:**

The experiments are technically sound but do not substantiate the central claim that “knowledge-tracing unlearning” differs meaningfully from data-tracing MU. The forgetting set is still composed of data samples corresponding to fine-grained classes, and the proposed loss modification merely adjusts the strength of GA. The evaluation treats the forgetting of fine-grained classes as evidence of knowledge-level unlearning. However, class labels are data annotations, not representations of knowledge. The reported QU and other metrics therefore measure sample-level performance degradation rather than the removal of internal knowledge. Without probing the model’s representation space or semantic behavior, the paper does not provide genuine evaluation at the knowledge level.

**Requested Changes:**

1. Provide a formal definition or representation of “knowledge” and specify what it means to “trace” and “forget” knowledge within a model.
2. Clarify the contradiction between the stated motivation (no access to training data) and the implementation (explicit use of class-level samples D^f).
3. The authors should tone down claims suggesting a new paradigm, as the method is essentially a small tweak to existing MU methods. It would also help to clearly separate the conceptual discussion (as a position paper) from algorithmic contributions, since the current version blurs the boundary between the two. While the final discussion section refers to the work as a “position paper”, the abstract and methodology present it as a concrete paradigm and algorithmic realization, leading to inconsistency in framing and expectations.

---

> ### Author Response · Authors · 2026-02-06
>
> **Weakness**
>
> **W1**:  We agree that formally defining “knowledge” in knowledge-tracing machine unlearning is inherently challenging, and the proper definition can vary substantially across domains and by different stakeholders. To address this concern, we have added a new section (Section 4: Challenges in knowledge-tracing FMU) that explicitly discusses these challenges.
> Importantly, our case study is intentionally a concrete instantiation rather than a full formalization of “knowledge-tracing” in general. We operationalize “knowledge” as fine-grained visual concepts (class categories). This choice provides a tractable starting point, but we do not claim that knowledge-tracing unlearning is limited to classes. In other domains, “knowledge” could reasonably correspond to entities/relations in a knowledge graph, factual statements, procedural skills, or higher-level semantic rules, to list a few, each of which would require different tracing mechanisms and success criteria.
>
> **W2**: Df does not correspond to the foundation model’s original training data; instead, it consists of unlearner-sourced proxy examples collected externally to instantiate a knowledge-level unlearning request. We emphasize that our paper does not claim that knowledge-tracing unlearning methods must be data-free, but rather that the original training data of foundation models are typically inaccessible. In our case study, we adopt a data-based instantiation for concreteness, while alternative data-free and/or training-free methods are also possible and left for future work.
>
> **W3**: The proposed method is specifically designed for knowledge-tracing machine unlearning, whose major difference from data-tracing unlearning is the lack of access to the target model’s training data. As a result, we also do not have a retention set — for data-tracing machine unlearning, the training data excluding the forgetting set naturally constitute the retention set. To address these data-scarcity challenges in knowledge-tracking unlearning, we add extensive regularization to the optimization objective, including the KL divergence and the hinge loss. The hinge loss is similar in spirit to NPO (Zhang et al., 2024); both are “a stabilization trick” that modify gradient ascent to mitigate catastrophic collapse. Meanwhile, the hinge loss has a regularization effect by preventing over-forgetting, which is especially important due to our lack of the retention set. Finally, we note that the additional KL regularization terms give rise to significant gains in the experiments.
>
> **W4**: We have clarified in the revision that “relying on data” does not imply access to the original training data.
>
> **W5**: We agree that this work is a position paper, and it is not intended to introduce technical novelty (although our algorithm is actually novel as we tailored it to the new problem setup).The primary contribution lies in reframing machine unlearning for foundation models from a data-tracing paradigm to a knowledge-tracing one, motivated by practical constraints and cognitive insights. The technical case study is included as an exemplar realization to demonstrate how such knowledge-level unlearning requests can be instantiated in practice, rather than as a claim of algorithmic novelty. We believe this conceptual shift and its concrete instantiation provide a useful foundation for future research on unlearning beyond data deletion.
>
> **Requested Changes**
>
> **R1**:  Please see the answer to the first weakness.
>
> **R2**:  Please see the answer to the second weakness.
>
> **R3**: We thank the reviewer for this important feedback. We would like to clarify that our work does not claim to introduce a novel unlearning algorithm. Rather, by referring to a “new paradigm,” we meant a shift in research focus: we advocate for moving from Data-Tracing Machine Unlearning, which targets the removal of specific training samples, to Knowledge-Tracing Machine Unlearning, which aims to erase concepts or capabilities—an issue that is particularly salient for foundation models. The method presented in the paper is intended solely as a case study to illustrate the feasibility of this perspective. It is not meant to represent a definitive solution. Instead, it serves as an initial exemplar instantiation of the proposed Knowledge-Tracing paradigm, with the goal of stimulating further discussion and future research in this direction.

---

### Review · Reviewer_QqMd · 2025-11-18

**Summary Of Contributions:**

Machine unlearning methods seek to remove the influence of a user's data on machine learning model predictions/outputs. With the recent emergence of foundation models, machine unlearning techniques have been developed for these models as well. This position paper aims to motivate the development of methodology to enable a foundation model to forget "knowledge", rather than remove the influence of individual data points. The paper provides motivation for this shift in focus, and demonstrates applying existing machine unlearning methods to a knowledge unlearning task, specifically, unlearning specific labels in CLIP while retaining broad categories.

The demo is an interesting proof-of-concept -- the proposed method seems effective in achieving low prediction error on the "forgotten knowledge" and high prediction error elsewhere, and in general it is interesting to observe that existing unlearning techniques can be leveraged for more general tasks.

In terms of weaknesses, as this is a position paper, I will mainly describe areas where I feel the argument is weak. The central claim of the paper, to my understanding, is that "Enabling knowledge unlearning in foundation models is an unstudied but important problem." I think that the paper (a) lacks clarity in defining knowledge unlearning, and (b) does not sufficiently motivate the importance of this problem.

**Audience:**

Yes

**Audience Explanation:**

If the authors provide clearer and more detailed justification, this could be an interesting open problem.

Some additional, minor issues:
- In the last sentence of the first paragraph, the author's last name should not be capitalized, and the first clause is incomplete ("indispensible part" of what?).
- Figure 1 is very difficult to interpret. What is the top arrow coming from? What are the thought bubbles representing? The caption is completely uninformative about the figure.

**Claims And Evidence:**

No

**Claims Explanation:**

First, the paper does not well-address the fact that "knowledge" is difficult to operationalize. The proposed knowledge-tracing unlearning interface would allow stakeholders to use "high-level semantic descriptions" to request knowledge to be removed. This is underdefined -- what constitutes successfully forgetting that knowledge? What are the boundaries of the knowledge (if you remove knowledge about a public figure, do you also remove knowledge about things that person has done)? What if the stakeholder's description is in any way vague about these things? The example task in Section 4 avoids this issue by addressing a much more specific problem: knowledge here is a categorical label that the model should not output. Although the authors mention these challenges briefly in Section 4.3, I argue that this is a prohibitive barrier to making progress on "knowledge-tracing unlearning" / making the position paper "actionable" as the authors claim. The problem requires clearer definitions and scoping. In service of clarity, perhaps formalizing as much of this open problem as possible would be helpful in inspiring future research and discussion.

Second, the paper needs a stronger argument for why enabling knowledge-tracing unlearning is an important problem.

The paper claims that many stakeholders should be able to make knowledge unlearning requests, namely regulators, external individuals and organizations, and internal developers.
- My understanding is that the initial push for machine unlearning research was based on privacy regulation requiring model developers to allow a user to remove her data's influence from the model, affirming her ownership of her personal data. In the knowledge unlearning case, who has the right to knowledge, and where do these rights begin and end? Are there ongoing cases or legal discourse indicating that regulators would like to enable model forgetting? Even if regulators eventually decide to require foundation models to "forget knowledge", without knowing what regulation will exist in the future, what objectives should we use for building knowledge unlearning methods?
- For external individuals and organizations, what incentive would company C have to respond to unlearning requests? If the incentive is financial -- these external individuals and organizations could pay to remove knowledge -- are there ethical considerations? If the incentive is to make the model more accurate (e.g. forgetting misinformation or mislabeled data), why should explicit unlearning/forgetting be the solution? If the incentive is to remove outdated knowledge (such as a discontinued brand in the authors' example), why should that knowledge be forgotten, rather than the model retaining that information as well as the fact that the knowledge is outdated?
- For the model developers themselves, I think the argument should be that knowledge-tracing will improve the performance (broadly speaking) of the model. However, I do not find the paper's arguments for this convincing. For example, authors claim at the bottom of page 2 that "FMs can likely benefit from this unlearning process by freeing up model capacities". Humans get to choose which information is important and which to abstract away, but the FMs here must forget knowledge specified by a user, which might not be chosen with parsimony / abstraction in mind. I don't have intuition is that this would likely free up capacity (unless the unlearning is very substantial), so this claim requires further justification.

For some of these questions I have some intuition what the answers may be; however, I think it is important that the paper explicitly grapples with these issues.

**Requested Changes:**

I recommend that the authors improve the clarity of what the proposed open problem is: what, formally, does it mean to be successful at knowledge-tracing unlearning. I also recommend that the authors add stronger arguments for why knowledge-tracing unlearning should be studied.

---

> ### Author Response · Authors · 2026-02-06
>
> **Weakness**
>
> **W1**:  We appreciate the reviewer’s insightful critique regarding the difficulty of operationalizing "knowledge" and defining its boundaries. These are indeed the core challenges that distinguish knowledge-tracing machine unlearning from traditional data-tracing unlearning. We have added a new section (Section 4: Challenges in knowledge-tracing FMU) that explicitly discusses these challenges.
>
> **W2.1**: We thank the reviewer for raising these important questions and hints for answering them. We clarify that knowledge-tracing unlearning is not grounded in any ownership of facts, as in data unlearning, but in accountability for model behavior. While data-tracing unlearning is motivated by privacy and data ownership, knowledge-tracing unlearning arises from the need to mitigate harmful, unsafe, or legally risky behaviors exhibited by foundation models. We agree that there is currently no explicit regulation mandating “knowledge forgetting.” However, generative AI systems are already subject to capability-level expectations (e.g., not producing unsafe or defamatory outputs). Accordingly, we propose knowledge-tracing unlearning as a regulation-agnostic technical primitive, with a concrete objective: reducing the likelihood of the targeted concept or capability, while robustly preserving the model's general knowledge and utility. This formulation provides a well-defined research objective that remains meaningful regardless of how future regulatory standards evolve.
>
> **W2.2**:  The primary motivation for a company to respond to knowledge-based unlearning requests is risk and liability mitigation. This includes avoiding potential litigation related to defamation, intellectual property infringement, or safety negligence. We explicitly reject the “pay-to-remove” premise. Knowledge unlearning should instead be viewed as a rights-based compliance mechanism, analogous to the “Right to be Forgotten,” where stakeholders demand the removal of harmful or improper content to enforce legal or safety obligations.
>
> We further argue that explicit knowledge unlearning is necessary and cannot be reliably replaced by simple updates, fine-tuning, or metadata tagging. Large foundation models are known to suffer from knowledge interference, where outdated or incorrect information continues to coexist with newer knowledge in overlapping representational subspaces. As a result, merely labeling or de-emphasizing such information often leaves residual “ghost weights” that can resurface under distribution shift or adversarial prompting. In contrast, structurally erasing the targeted knowledge through unlearning provides a more robust safeguard against the reactivation of harmful capabilities.
>
> **W2.3**: Foundation models often suffer from parametric interference, where conflicting or obsolete knowledge competes for overlapping representational subspaces, which can manifest as unstable decision boundaries or hallucinated outputs. From this perspective, knowledge-tracing unlearning can be viewed as resolving internal representational conflicts. For example, removing toxic reasoning patterns or outdated facts may reduce noise around the decision boundary, improving the signal-to-noise ratio of the remaining knowledge and enabling more effective use of existing parameters, without changing model size. In the proposed paradigm, the developer acts as the active agent of abstraction, using knowledge unlearning as a surgical tool to prune non-parsimonious branches of knowledge.
>
>
> **Requested Changes**
>
> **R1**: Thank you for pointing this out. We revised it in the updated version.
>
> **R2**: We revised Figure 1 and its caption in the updated version.

---

### Review · Reviewer_8dqg · 2026-01-23

**Summary Of Contributions:**

This paper aims to elevate machine unlearning from the data-level to the knowledge-level, providing a rationale for machine unlearning in foundation models and clarifying how it differs from traditional data-level approaches. The authors further attempt to connect unlearning in foundation models to human forgetting, drawing inspiration from cognitive psychology to address the current ambiguity in defining machine unlearning for foundation models. To support their claims, the authors design experiments using hierarchical knowledge structures, selectively removing certain knowledge nodes to simulate knowledge-trace unlearning. This approach is used to validate existing methods and compare them to the proposed technique.

Here are strengths and weaknesses:

Strengths:
1. Clear position of the paper, pointing out the difference from data-tracing unlearning as the foundation model acts differently from the standard discriminative model. Moreover, the paper designed the experiments to echo their research problem.

2. The connection between human forgetting from the cognitive and psychology perspective is interesting, supporting the importance of FMU.

Weaknesses:
1. While the paper claims a novel perspective, similar ideas have been implicitly discussed in prior works. For example, class unlearning evaluation setups closely resemble the approach proposed here, and hierarchical datasets have already been used to assess machine unlearning methods in previous research.

2. The writing, particularly regarding the proposed method, lacks clarity. Key details are not explicitly presented in the main text; for instance, the final objective function only appears in Table 10 of the appendix. Additionally, the role of KL divergence in the proposed method is not substantiated through ablation studies, leaving its importance unvalidated.

**Audience:**

Yes

**Audience Explanation:**

Machine unlearning is an emerging and increasingly important topic in AI. Given the rapid advancements in artificial intelligence, it is crucial for the research community to deepen its understanding of machine unlearning. This will help ensure responsible AI development and mitigate potential catastrophic consequences that may arise from improper use or retention of sensitive information in AI systems.

**Broader Impact Concerns:**

No concern on the ethical implication and do not need a broader impact statement.

**Claims And Evidence:**

Yes

**Claims Explanation:**

The authors designed a specific experimental setup to echo their claims; moreover, the experimental results show that the proposed algorithm achieved promising results compared to other methods.

**Requested Changes:**

1. The concept of knowledge-tracing has been studied in the cited works, like NPO, I do not know what the uniqueness pointed bring by the authors, e.g., the standard class unlearning covers the scenario the auother mentioned; moreover, there are multiple works that using hirerachical datasets as the experimental setup as the knowledge unlearning. A stronger justification, such as a clearly articulated new research question, would help clarify the novelty of this work.

2. The proposed method is unclear, is it just GA with hinge loss? or something more? and what does NHL stand for? The final objective is written in Table 10 but what is L_u? and for most of results, it is reported along with KL, what is the performance without KL?

3. The authors mentioned that Cha et al. is the concurrent work that applies the hinge loss to LLMs. But that works is published at ICLR 2025 (Apr 2025), Shouldn’t this be considered prior art rather than concurrent work? Even if it is concurrent, the authors should elaborate on the differences between their approach and Cha et al., and discuss the potential advantages of their proposed method.

4. In the section 4, the description associated to figure 3 seems to be inconsistent, there is no Oudi O1 images in figure 3, it might be easier to make the reader to understand the example with consistent illustrations.

5. Regarding the forgetting difficulty, could it be designed in a way the level of knowledge to-be-forgotten? For example, in figure 1, forgetting dog, terrier and boston terrier could be different difficulty. Exploring this aspect could provide deeper insights into the unlearning process.

---

> ### Author Response · Authors · 2026-02-06
> **Reply to the Weakness**
>
> **Weakness**
>
> **W1**: While similar ideas might “have been implicitly discussed in prior works,” our goal is to promote knowledge-tracing (vs. data-tracing) machine unlearning to the first-class citizen rather than leaving it as something implicit. Additionally, existing class unlearning setups typically rely on access to the model’s full original training data, where unlearning is performed by explicitly revisiting and modifying the same class samples used during training.  In contrast, our case study focuses on the foundation model setting, in which the original training data are often inaccessible, or even unknown.
> To the best of our knowledge, existing works have not explicitly leveraged hierarchical labels in their studies yet. In contrast, we use hierarchical knowledge to simulate unlearning requests at different levels of the knowledge granularity.
>
> **W2**: Thank you for the feedback! In the revised paper, we have moved the overall loss function from the appendix into the main text. Regarding the role of KL divergence, we examine it by a controlled experiment, whose results are in Table 2. Incorporating the KL term consistently improves performance, validating its importance in the proposed method.

---

> > ### Author Response · Authors · 2026-02-06
> > **Reply to the Requested Changes-1**
> >
> > **Requested Changes**
> >
> > **R1**: We want to clarify that NPO follows the paradigm of data-tracing machine unlearning, rather than the knowledge-tracing setting proposed in our work. Specifically, NPO first trains/tunes a language model on a corpus of fictitious data and then performs unlearning by explicitly removing a fixed portion of that training data. Hence, NPO’s unlearning follows the conventional data removal. Moreover, existing works have not explicitly leveraged hierarchical labels to simulate unlearning requests at different levels of visual knowledge granularity, to the best of our knowledge.
> >
> > **R2**: The proposed method is specifically designed for knowledge-tracing machine unlearning, whose major difference from data-tracing unlearning is the lack of access to the target model’s training data. As a result, we also do not have a retention set — for data-tracing machine unlearning, the training data excluding the forgetting set naturally constitute the retention set. To address these data-scarcity challenges in knowledge-tracking unlearning, we add extensive regularization to the optimization objective, including the KL divergence and the hinge loss.
> > Following the reviewer’s feedback, we have moved the optimization objective from the appendix to the main paper. It is now clear that it is “just GA with hinge loss” with a great deal. Indeed, the hinge loss offers a dual effect: GA over the forgetting set and aversion to over-forgetting without the use of a retention set. Besides, the two KL divergence terms in the objective provide additional regularization despite the lack of the retention set.
> > Thank you for catching “NHL” — it was unclear. In the revised paper, we have instead named the hinge loss as HGA, standing for hinged gradient ascent.
> >
> > **R3**:  Cha et al.’s work is indeed concurrent with ours. The main difference between the two is on the access of the retention set, as explained in our response to the previous question. Besides, technically, Cha et al. apply the hinge loss to softmax probabilities for LLMs, while we use it over image-text similarity scores for Vision-Language Models.
> > Finally, we would like to clarify that our primary contribution is fundamentally different from Cha et al,’s because this work is the advocacy for the knowledge-tracing machine unlearning paradigm for FMs, rather than a new unlearning algorithm per se.

---

> > > ### Author Response · Authors · 2026-02-06
> > > **Reply to the Requested Changes-2**
> > >
> > > **R4**: Thanks for your suggestion. We have updated it in the revised paper.
> > >
> > > **R5**: We thank the reviewer for this insightful suggestion. We agree that exploring the difficulty of forgetting across different levels of the knowledge granularity can provide deeper insights into the knowledge-tracing machine unlearning process.
> > >
> > > Unlearning general, coarse-grained concepts may pose a significant risk to model utility, as removing foundational nodes (e.g., "dog") disrupts shared feature representations and degrades general capabilities. In contrast, unlearning fine-grained concepts challenges the model's precision, requiring the surgical disentanglement of specific targets (e.g., "Boston terrier") from highly similar siblings to prevent over-forgetting.
> > >
> > > We have added a new experiment on unlearning coarse-grained classes (e.g., “Retriever” and “Setter”), which complements our original case study on fine-grained concepts. In this setting, the model is required to unlearn both the coarse-grained concept and all subordinate fine-grained concepts under that class. We observe that, when unlearning coarse-grained classes, balancing unlearning quality and model utility is more challenging than for fine-grained classes, as reflected by a lower Q–U metric for our proposed method.
> > >
> > > Importantly, the difficulty of unlearning is not determined solely by concept granularity but also by the model’s degree of memorization. For example, CLIP performs worse on coarse-grained categories than on fine-grained ones. As a result, for some methods, unlearning fine-grained concepts under a coarse-grained class can require greater effort than unlearning the coarse-grained concept itself, highlighting that unlearning difficulty depends jointly on concept granularity and the target model’s idiosyncrasies.
> > >
> > > | Method | $\mathcal{D}^f$ coarse $\downarrow$ | $\mathcal{D}^f$ fine $\downarrow$ | $\mathcal{D}^r$ coarse $\uparrow$ | $\mathcal{D}^r$ fine $\uparrow$ | Quality $\uparrow$ | Utility $\uparrow$ | Q-U $\uparrow$ | Zero-shot $\uparrow$ |
> > > | :--- | :---: | :---: | :---: | :---: | :---: | :---: | :---: | :---: |
> > > | Origin | 76.75 | 80.25  | 53.45 | 67.32 | -| - | - | 83.24 |
> > > | GA | 0.00 | 0.00 | 10.33 | 29.07 | 100.00 | 31.25 | 47.62 | 80.52 |
> > > | GDiff | 0.00 | 0.00 | 10.66 | 28.46 | 100.00 | 31.11 | 47.46 | 81.20 |
> > > | GA+KL | 0.00 | 0.25 | 42.44 | 40.84 | 99.84 | 70.03 | 82.32 | 82.30 |
> > > | Relabeling | 14.75 | 16.00 | 41.65 | 43.64 | 80.42 | 71.37 | 75.63 | 81.02 |
> > > | SaLUN | 7.25 | 29.50 | 51.69 | 59.30 | 76.90 | 92.40 | 83.94 | 82.55 |
> > > | ME+GD | 22.50 | 31.25 | 50.18 | 50.15 | 65.87 | 84.19 | 73.91 | 82.19 |
> > > | Task Vector | 8.75 | 28.00 | 55.45 | 63.70 | 76.85 | 99.18 | 86.60 | 82.64 |
> > > | NPO+KL | 8.25 | 24.50 | 55.03 | 63.03 | 79.36 | 98.29 | 87.82 | 83.14 |
> > > | **HGA+KL** | 10.50 | 8.00 | 52.35 | 60.09 | 88.18 | 93.60 | 90.81 | 82.99 |

---

### Decision · Action_Editor_5JH2 · 2026-04-08

**Recommendation:** Accept as is

**Audience:**

Yes

**Audience Explanation:**

The machine unlearning problem is interesting to the ML/AI community, especially those who work on large foundation models.

**Claims And Evidence:**

Yes

**Claims Explanation:**

The paper provides empirical evidence and experimental results to support the claims.